# Data-Driven Power Prediction for Proton Exchange Membrane Fuel Cell Reactor Systems

**DOI:** 10.3390/s24186120

**Published:** 2024-09-22

**Authors:** Shuai He, Xuejing Wu, Zexu Bai, Jiyao Zhang, Shinee Lou, Guoqing Mu

**Affiliations:** 1School of Mechanical & Automotive Engineering, Qingdao University of Technology, Qingdao 266520, China; 2Qingdao Chuangqi Xinde New Energy Technology Co., Ltd., Qingdao 266100, China; 3School of Chemical Engineering and Technology, Tianjin University, Tianjin 300072, China; 4School of Information and Control Engineering, Qingdao University of Technology, Qingdao 266520, China

**Keywords:** PEMFC stacks, power prediction, data-driven, BP-AdaBoost

## Abstract

Enhancing high-performance proton exchange membrane fuel cell (PEMFC) technology is crucial for the widespread adoption of hydrogen energy, a leading renewable resource. In this research, we introduce an innovative and cost-effective data-driven approach using the BP-AdaBoost algorithm to accurately predict the power output of hydrogen fuel cell stacks. The algorithm’s effectiveness was validated with experimental data obtained from an advanced fuel cell testing platform, where the predicted power outputs closely matched the actual results. Our findings demonstrate that the BP-AdaBoost algorithm achieved lower RMSE and MAE, along with higher R^2^, compared to other models, such as Partial Least Squares Regression (PLS), Support Vector Machine (SVM), and back propagation (BP) neural networks, when predicting power output for electric stacks of the same type. However, the algorithm’s performance decreased when applied to electric stacks with varying material compositions, highlighting the need for more sophisticated models to handle such diversity. These results underscore the potential of the BP-AdaBoost algorithm to improve PEMFC efficiency while also emphasizing the necessity for further research to develop models capable of accurately predicting power output across different types of PEMFC stacks.

## 1. Introduction

In recent years, the exigent challenges posed by environmental pollution and energy shortages have instigated a global transition towards sustainable and eco-friendly energy solutions [1]. Among these, hydrogen energy has emerged as a preeminent contender, endowed with the potential to revolutionize energy consumption across diverse sectors. Whether it is for energizing vehicles and portable devices or furnishing a reliable source for stationary power generation, hydrogen’s high energy density and clean-burning characteristics render it an ideal candidate to satisfy future energy demands [2].

The hydrogen fuel cell constitutes one of the crucial technologies indispensable for the extensive application of hydrogen energy, particularly the proton exchange membrane fuel cell (PEMFC) [3]. Hydrogen at the anode liberates electrons for the oxidation reaction, while oxygen at the cathode acquires electrons for the reduction reaction. In recent years, PEMFCs have gained attention for their environmental friendliness, high energy conversion efficiency, low noise, and low operating temperature. However, they also have certain limitations, such as their high cost and a slow electrochemical reaction rate. Accurate power prediction is crucial for optimizing energy distribution between the fuel cell and other energy sources in a hybrid system, thereby enhancing overall system performance. It also contributes to better fuel utilization, reduced fuel consumption, and lower operating costs [4]. PEMFCs are known for their low operating temperature, lightweight design, high energy density, sensitive reactions, and lack of pollution. As a result, PEMFCs have the potential to be widely utilized in small-, medium-, and large-scale portable devices as well as general-purpose technology for electric vehicles [5].

Where the performance of the fuel cell is monitored through cell voltage measurements, this process uses a Cell Voltage Monitoring (CVM) [6] system to gather data on the operating conditions of the PEMFC. In an ideal PEMFC stack, each cell is under the same conditions and the overall performance of the stack is the sum of the same outputs of the individual cells, but in reality, due to manufacturing differences in components, stack structure, and degradation during use, individual cells will exhibit some differences in performance, depending on the specific operating conditions, affecting the overall stack performance [7]. The CVM compares the minimum and maximum values of each battery voltage and triggers an alarm if the voltage is outside this range, allowing the battery stack to operate at a higher power level if under-voltage is actively detected. The challenge with this direct experimental method is its cost. The complexity and expense come from the need to monitor each cell within the stack separately. For accurate power prediction, the system might require individual monitoring components for each cell, such as isolated analog-to-digital converters (ADCs) and their power supplies. For example, the cost of producing these components for a set of 3000 units exceeded $100, highlighting the high expenses involved in the experimental measurement of PEMFC power [8]. This cost issue raises concerns about the commercial viability of hydrogen fuel cells, considering both their lifespan and the investment required for their monitoring systems [9]. The battery voltage monitoring system is a critical component of PEMFC. Our data-driven method predicts the power of PEMFC using process variables, which not only supports real-time system monitoring but also accurately determines the relationship between power and process variables, and aids in monitoring hydrogen energy consumption. The utilization of model-driven methods necessitates a thorough comprehension of the internal parameters and mechanisms of PEMFC. Xie [10] demonstrated that more accurate predictions of PEMFC performance can be achieved by constructing a 3D agglomerate sub-model of the catalyst layer. However, due to variations in process parameters and physicochemical phenomena across different types of fuel cells, this approach cannot be universally applied. Additionally, the use of a simplified physical model may lead to a loss of critical information, reducing the accuracy of predictions. So, there is a great difficulty in predicting some properties of PEMFC by model-driven methods [11].

There is a need to propose a new model to accurately predict the overall performance of PEMFC and to adapt the model to improve the operational efficiency. The data-driven approach bypasses the complexities of fuel cell degradation mechanisms by using machine learning to create a black-box model based on experimental aging data. Currently, machine learning, particularly deep learning, is increasingly applied in the field of fuel cells. For example, Meng et al. [12] used the Transformer model to predict fuel cell performance decline. The model’s unique self-attention structure and masking mechanism were utilized to recover reversible voltage loss signals as input, enabling the prediction of accelerated fuel cell degradation. Liu et al. [13] developed a method using Long Short-Term Memory (LSTM) Recurrent Neural Networks (RNN) to predict the remaining useful life (RUL) of proton exchange membrane fuel cells (PEMFC). This method employs regular interval sampling and locally weighted scatter plot smoothing for data reconstruction and smoothing, allowing for quick and accurate RUL predictions. Long et al. [14] proposed a GRU-based RUL prediction method for hydrogen fuel cells. Studies showed that, at a starting point of 650 h, the accuracy of the GRU model was several times higher than that of the BP Neural Network (BPNN) model. Whether starting at 650 or 850 h, the GRU model outperformed the BPNN model in terms of RMSE, MAPE, and R^2^. Although the prediction accuracy of the GRU model is close to that of the LSTM model, the GRU converges faster, making it a more advantageous method for RUL prediction compared to BPNN and LSTM. A data-driven approach with ridge regression and convolutional neural networks was proposed [15] to predict the power of real PEMFC. Liu et al. [16] introduced a Markov chain model for power prediction and incorporated a reinforcement learning technique to accelerate computational convergence. Their findings show that the proposed model can dynamically update power demand based on current driving conditions, thereby improving energy management, enhancing fuel economy, and boosting real-time performance. Krastev et al. [17] introduced techniques to forecast the electrochemical and power efficiency of microbial fuel cells using the Boltzmann approach. Weida Wang [18] utilized the unscented Kalman filter (UKF) to develop a prediction method for peak battery power, based on the State of Charge (SOC) and battery modeling. This approach enhances peak power prediction and battery management for Hybrid Electric Vehicles (HEVs). Lastly, a novel power prediction index for fuel cell hybrid systems, utilizing the State of Power (SOP) algorithm, has been introduced. This index enhances strategic energy management and improves the accuracy of power predictions for the battery. Additionally, it includes a method for ensuring the safe operation of energy systems [19]. Despite the advancements in existing methodologies, they fail to adequately account for the correlation between the operational variables of the power reactor system and the reactor’s power output. This limitation hampers the development of reliable data-driven models. A persistent issue across studies is the insufficient integration of operational variables with power output in predictive models. Addressing this gap remains a critical avenue for future research, essential for building more dependable models for predicting fuel cell power output.

Given the cost implications and limitations of traditional models, there has been growing interest in exploring data-driven methods for power prediction as a way to reduce expenses. In hydrogen fuel cell research, a range of data-driven models has been developed to predict and enhance the power output and efficiency of fuel cells, representing significant progress in the pursuit of sustainable and efficient energy solutions [20].

To facilitate readers clearly understanding the article, the innovative points of this paper are as follows:(1)This research showcases a data-driven modeling approach utilizing the BP-AdaBoost algorithm, marking the first time this method has been used to predict power output from process variables; it also helps to monitor hydrogen energy consumption.(2)The BP-AdaBoost method proposed in this paper can achieve a good prediction effect in the same type of hydrogen fuel reactor and provides a feasible scheme for further research.

The specific structure of the paper is as follows: the second part is the experimental setting and data collection methods, the third part is the overview of the model and evaluation methods, the fourth part analyzes the experimental results in detail, and the last part summarizes the research conclusions and prospects.

## 2. Experimental Setup and Data Acquisition

The PEMFC stack generates electricity through a chemical reaction, with each cell containing two electrodes: an anode and a cathode. At the anode, hydrogen undergoes oxidation, releasing electrons and producing hydrogen ions. Meanwhile, at the cathode, oxygen accepts the electrons and combines with the hydrogen ions to form water. This process generates a flow of electrons, creating an electric current that powers external equipment, while producing only water as a clean byproduct. It is important to note that a single fuel cell produces a relatively small amount of electricity, which is why multiple cells are combined into a stack to generate sufficient power for practical applications, as illustrated in Figure 1b.

In this research, the PEMFC stacks were evaluated using a sophisticated fuel cell testing system, the RIGOR-RG23010 is manufactured in Dalian, China by Dalian Rigor New Energy Technology Co., Ltd., Dalian, China, capable of testing up to 60 kW. The setup involved automated connections for hydrogen, air, and cooling water inlets/outlets to the testing platform, as shown in Figure 1a,b in the schematic diagram.

Throughout the evaluation, 27 process variables were continuously monitored on a second-by-second basis through integrated sensors, with the specifics of these variables detailed in Table 1.

In this study, two PEMFC stacks, referred to as stack A and stack B, were evaluated. Each stack underwent testing across a range of loading conditions, with all 27 process variables being meticulously recorded.

In the experiment, it was possible to directly measure both the overall voltage (U) and current (I), enabling the calculation of power. This calculated power could then be compared with values predicted by data-driven methods.

## 3. PEMFC Power Prediction Model Construction

### 3.1. BP-AdaBoost Based Power Prediction Method for Power Stacks

The BP-AdaBoost model [21] combines the AdaBoost algorithm with a back propagation (BP) neural network to form a robust predictive framework. In this model, the BP neural network [22], serves as the weak learner, laying the foundation for BP-AdaBoost’s predictive capabilities. This approach leverages BP networks as weak learners to efficiently extract data features by optimizing the sample node weights and weak predictors [23]. Each training iteration begins by initializing the sample weights, followed by fine-tuning the weak learners based on the current dataset’s sample weights. Through this iterative process of weight optimization and integrating outputs from multiple weak learners, a strong learner is formed as an ensemble of weak learners [24,25].

The flowchart of the algorithm is shown in Figure 2.

The major calculation steps are as follows:

(1) The m samples of hydrogen fuel cells are used as the training dataset:(1)$$D=\left\{\left(x_{1},y_{1}\right),\left(x_{2},y_{2}\right),\ldots ,\left(x_{m},y_{m}\right)\right\}$$
where *x_i_* is the process variable of the hydrogen fuel cell, and *y_i_* is the true power value of the PEMFC Stacks.

Initialize the weight distribution of the training data such that, at the outset, each sample is assigned an equal weight using the AdaBoost algorithm.
(2)$$\omega_{1i}=\frac{1}{m},i=1,2,\ldots ,m$$
where *ω*_1*i*_ is the initial weight size of each training sample.

(2) Iterative updating:

The loop runs for T iterations, with each iteration represented by learner t. Within the loop, AdaBoost is primarily employed to train the BP neural network by calculating its error rate and updating the weights accordingly to train the next BP neural network based on this error. Ultimately, all the trained BP neural networks are combined to form a strong learner, which is the BP-AdaBoost model proposed in this paper. The loop follows the steps outlined below, calculated as follows: 

(a) Training BP neural networks: In this paper, the weak learner uses a BP neural network, and the weights of the current samples are utilized to train the weak learner. During the training process, the network parameters are updated by the back propagation algorithm so that the network can approximate the experimental value of the samples. For *t* = 1, 2, …, T. And the calculation formula is:(3)$$h_{t}\left(x\right)=kx_{i}$$
where *k* is the ratio of power to process variable.

(b) Calculate the maximum error of the weak learner: Train the weak learner and predict the samples, then compute the error as the sum of the weights of the misclassified samples. Based on this error, update the BP neural network’s weights—the lower the error, the higher the weights assigned to the BP neural network. This process is performed on the training set using the following formula:(4)$$E_{t}=\max\left|y_{i}-h_{t}\left(x_{i}\right)\right|,i=1,2\ldots m$$where *h_t_(x_i_)* denotes the power value predicted by the BP neural network and *y_i_* is the true power value.

(c) Based on the maximum error *E_t_* in *h_t_* from the previous step, calculate the relative error of the BP neural network for each sample using the mean error.
(5)$$e_{ti}=\frac{{\left(y_{i}-h_{t}\left(x_{i}\right)\right)}^{2}}{E_{t}^{2}},i=1,2,\ldots ,m$$

(d) Based on the obtained sample relative error *e_ti_* the error rate of the BP neural network is finally obtained:(6)$$e_{t}=\sum_{i=1}^{m}\omega_{ti}e_{ti}$$
i.e., the sum of its products of the weights and errors of all the samples in the dataset, where *ω_ti_* is the weight at the *t*-th iteration.

(e) With the error rate, the weight coefficients *α_t_* of the BP neural network can be computed. The formula is as follows:(7)$$\alpha_{t}=\frac{e_{t}}{1-e_{t}}$$

(f) Update the sample weights.

After obtaining the weights of the BP neural network, the sample weights are adjusted based on both the neural network weights and the prediction results. If the data samples from the electric stack are accurate, their weights decrease. Subsequently, the BP neural network is trained using the training set with weights *ω_ti_*; the sample weights are then updated according to the weight coefficients *α_t_* of the BP neural network. The updated sample weights *ω_t_*′ are used as the weights for the next round of iterations, continuing the training of the subsequent BP neural network. The updated formula for the sample weights is as follows:(8)$$\omega_{t+1,i}=\frac{\omega_{ti}}{Z_{t}}\alpha_{t}^{1-e_{ti}}$$
where *Z_t_* is the normalization factor:(9)$$Z_{t}=\sum_{i=1}^{m}{\omega_{t}}^{\prime }\alpha_{t}^{1-e_{ti}}$$

(g) If *i* < *t*, then make *i* = *i* + 1 and return (d); otherwise, perform step (3).

(3) Normalized sample weights: normalize the sample weights so that the sum of the weights is 1.

(4) Portfolio Strategy.

Ending the T rounds of iterations, the final strong learner is as follows:(10)$$H\left(x\right)=h_{t^{*}}(x)$$
where *h_t*_*(*x*) is of all *ln 1/α_k_*, *t = 1*, *2*, …, *T*, and the median multiplied by the corresponding ordinal number *t** for the corresponding BP learner. 

### 3.2. Model Evaluation Indicators

The main evaluation indicators of the model are the following:

(1) Root Mean Square Error (RMSE): RMSE is a widely used metric for evaluating regression models, quantifying the deviation between predicted and true values. It measures the magnitude of prediction errors and is particularly sensitive to larger errors. A smaller RMSE indicates higher prediction accuracy of the model. The RMSE is calculated using the following formula:(11)$$RMSE=\sqrt{\frac{1}{m}\sum_{i=1}^{m}{\left(y_{i}-\hat{y}\right)}^{2}}$$

(2) Mean Absolute Error (MAE): MAE is a commonly used metric for evaluating regression models. It measures the average magnitude of the differences between predicted and true values by calculating the mean of the absolute differences. A smaller MAE indicates that the model’s average prediction error is low. The MAE is calculated using the following formula:(12)$$MAE=\frac{1}{m}\sum_{i=1}^{m}\left|\left(y_{i}-{\hat{y}}_{i}\right)\right|$$

(3) Coefficient of Determination (R^2^): It is a statistical indicator used to evaluate the effectiveness of the model by comparing the variance between the actual observations and the predicted values. A value of R^2^ close to one indicates a good fit, meaning the model explains a high proportion of the variance in the data. Conversely, a negative R^2^ suggests a poor model fit. The formula for R^2^ is:(13)$$R^{2}=1-\frac{\sum_{i=1}^{m}{\left(y_{i}-{\hat{y}}_{i}\right)}^{2}}{\sum_{i=1}^{m}{\left(y_{i}-\bar{y}\right)}^{2}}$$where *y_i_* denotes the true power value, *y* denotes the average of the true power values, *ŷ_i_* denotes the predicted power value, and m shows the total number of samples.

The combined use of RMSE, MAE, and R^2^ provides a comprehensive evaluation of a model’s performance from multiple perspectives. RMSE and MAE assess the magnitude of prediction errors, while R^2^ measures the model’s fit to the data. By considering these metrics together, we gain a thorough understanding of the model’s strengths and weaknesses, leading to a more objective and complete assessment of the results. This approach helps in selecting the most suitable model for prediction and analysis.

## 4. Discussion and Analysis of Results

The first experiment predicted the power output of PEMFC stack A, while the second experiment focused on predicting the power output of PEMFC stack B. The primary distinction between these stacks is the different catalysts used in their Catalyst-Coated Membranes (CCM). For both experiments, 70% of the data from one of the training electric stacks was used for training, and the remaining 30% was reserved for validation to fine-tune model parameters. In the first experiment, a set of stacks A with the same material composition as the training stacks was used as the test set. The subsequent experiment tests the method’s resilience by employing a different material composition electric stack B for testing. The training set remains with the power and process variables from electric stack A, with the test set incorporating analogous variables from electric stack B. Data were collected consistently at one data point per second, with select data points highlighted in visual representations to illustrate power trends within the PEMFC stacks.

### 4.1. Comparative Analysis of Power Prediction for the Same PEMFC Stacks

As shown in Table 2, for the same PEMFC stack, the training set metrics (MSE, MAE, and R^2^) of the PLS, SVM, BP, and BP-AdaBoost algorithms, as well as the direct multiplication of U and I, exceed the performance metrics of the test set. This result indicates that the data-driven approach effectively captures the relationship between process variables and power output during training, validating the effectiveness of the model training. The predictive power of the model is significantly enhanced due to the ensemble nature of BP-AdaBoost, which combines multiple weak learners. The evaluation metrics show the smallest differences compared to those of the U and I multiplication method, with R^2^ values closer to one, demonstrating that the model fits well and has good generalization ability. While the PLS method performs adequately, it does not match the error metrics and R^2^ values of the U and I multiplication and BP-AdaBoost methods. The SVM and BP methods, with higher error metrics and relatively lower R^2^ values, show weaker prediction capabilities. Among these methods, BP-AdaBoost delivers optimal prediction accuracy in both the training and test sets. However, it is essential to note that while the power output comparison involves U and I multiplication, this approach introduces minor errors. Despite this, the high cost and impracticality of gathering real-time currents and voltages render this method unsuitable for certain industrial settings.

In Figure 3, the actual power output from the PEMFC stacks is depicted by a blue line, while the model’s predicted power output is represented by a red line. This figure effectively highlights the discrepancies between actual and predicted power outputs for the PLS and SVM models in both the training and test datasets. Conversely, BP and BP-AdaBoost exhibit comparable accuracy in the training set, but BP-AdaBoost’s predictions on the test set are markedly superior to those of BP. Figure 3 also illustrates that while PLS, SVM, and BP models capture the general trend in both the training and test sets, their differences from the actual values are more significant compared to the BP-AdaBoost model, indicating that while these models can capture general trends, they struggle with accurately predicting specific values and exhibit poor generalization ability.

Furthermore, the power calculations derived from BP-AdaBoost’s predictions, as shown in Figure 3g,h, demonstrate a strong correlation with the results obtained through U and I multiplication, further validating the model’s reliability. This alignment underscores BP-AdaBoost’s effectiveness, which is further emphasized in Figure 3k,l, where the method exhibits the lowest absolute error among various data-driven approaches. The near-zero deviation in its predictions confirms the accuracy of BP-AdaBoost, as demonstrated in this study. This consistency across different evaluations highlights the robustness and precision of the BP-AdaBoost model in predicting power output.

### 4.2. Comparative Results Analysis of Power Prediction of Various PEMFC Stacks

The experiments indicate that the BP-AdaBoost model demonstrates strong predictive ability for the same type of PEMFC. To verify its feasibility, we extended our study to evaluate its power prediction performance across different types of PEMFC. As shown in Table 3, the prediction accuracy of the BP-AdaBoost method is lower for different types of PEMFC compared to the same type. This discrepancy is likely due to variations in the internal structure of the PEMFC stacks, such as differences in catalyst material.

For different types of PEMFC stacks, a significant performance gap is observed between the training set and test set for the PLS, SVM, BP, and BP-AdaBoost methods, indicating a decrease in the generalization ability of these models across different stacks. While BP-AdaBoost outperforms other methods in both the training and test sets, the overall performance of the PLS algorithm is not significantly different from BP-AdaBoost. The PLS method, which is prone to overfitting when selecting different numbers of principal components, is essentially a linear model and struggles to capture strong nonlinear relationships, thereby affecting model performance. Consequently, in practical situations, BP-AdaBoost is more effective than PLS for predicting hydrogen fuel cells. However, despite certain advantages, the overall predictive ability of BP-AdaBoost is still limited. In the test set, the RMSE and MAE are relatively high, and the R^2^ value is relatively low. Moreover, the evaluation metrics on the test set are far worse than those on the training set, indicating that the method does not generalize well to unseen data and cannot provide highly accurate predictions. Therefore, the BP-AdaBoost method cannot be generalized to different types of stacks by predicting the same type of stacks. Further research is necessary to develop more accurate prediction methods for PEMFC stacks with different compositions.

Figure 4 provides a visual comparison of the fitted and predicted values of various methods against the actual experimental data. As seen in Figure 4a–f,i,j, none of these four models can ensure that the actual power of different types of PEMFC matches the predicted power in the prediction set, with the gap between predicted and actual values being too large, leading to poor curve fitting and prediction ability. However, as shown in Figure 4k,l, the BP-AdaBoost method has slightly better prediction ability than the other three methods. It most closely captures the relationship between process variables and power output, better than the other methods. Nonetheless, the overall accuracy of power prediction using this method lags significantly behind that of multiplying U and I, indicating that the superiority of this approach is not conclusively proven for different types of PEMFC stacks. Therefore, it becomes essential to explore and develop more optimal methods that can accurately predict power output across various types of PEMFC stacks.

## 5. Conclusions

This research introduces several data-driven approaches for predicting power output in PEMFC stacks, capable of directly utilizing process variables for accurate power prediction. Experimental data confirm that the BP-AdaBoost algorithm achieves superior predictive accuracy for the same type of PEMFC stack, with an R^2^ of 0.9991, outperforming the PLS model, SVM model, and BP neural network. This is because AdaBoost synthesizes the advantages of multiple weak learners, such as through weighted combinations, to obtain a more robust model.

However, when tested on various types of fuel cell stacks, all algorithms, including BP-AdaBoost, failed to accurately predict peak power output values. Among them, the highest R^2^ value of the BP-AdaBoost prediction model in these cases was only 0.6637, showing significant degradation in performance. This indicates the limitations of the proposed method when applied to different types of PEMFC stacks, primarily due to differences in material composition, especially the different catalyst materials used in the catalytic layer, which affects the model’s predictions.

Overall, while the BP-AdaBoost method demonstrates strong predictive performance for the same type of PEMFC reactor, it has limitations when applied to different types of reactors. This difference underscores the need for further research to develop more accurate prediction methods that can optimize various PEMFC components. As differences in material composition increase, more complex models may be required for deeper analysis, such as Bayesian dynamic modeling, in our research. Future research should focus on refining these models to improve their applicability and accuracy across different types of PEMFCs, and on addressing noise in industrial applications, ultimately enhancing the efficiency and reliability of hydrogen fuel cells.

## Figures and Tables

**Figure 1 sensors-24-06120-f001:**
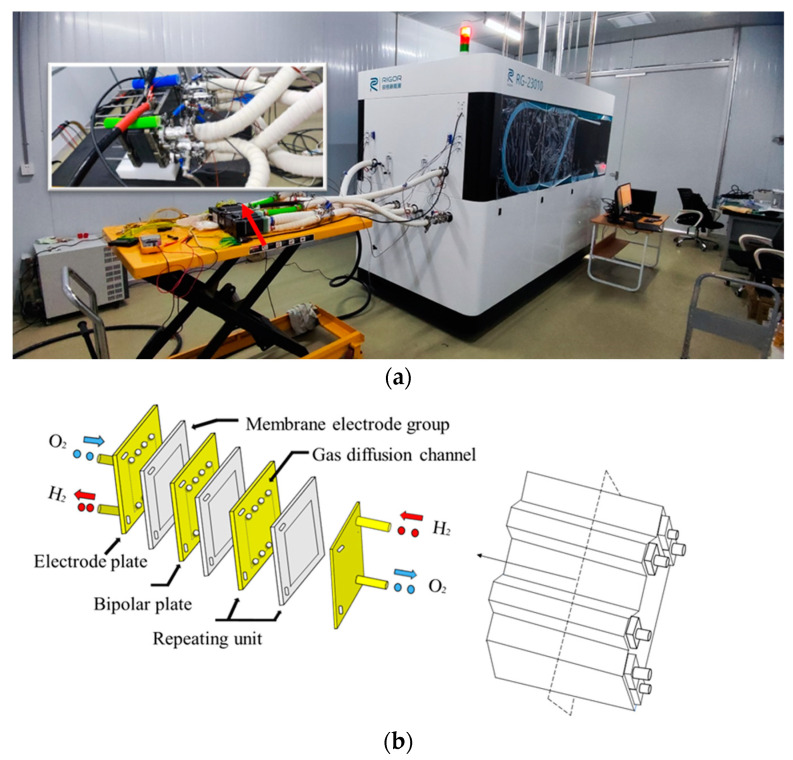
PEMFC Stack Test Platform: (**a**) exterior appearance, (**b**) schematic diagram.

**Figure 2 sensors-24-06120-f002:**
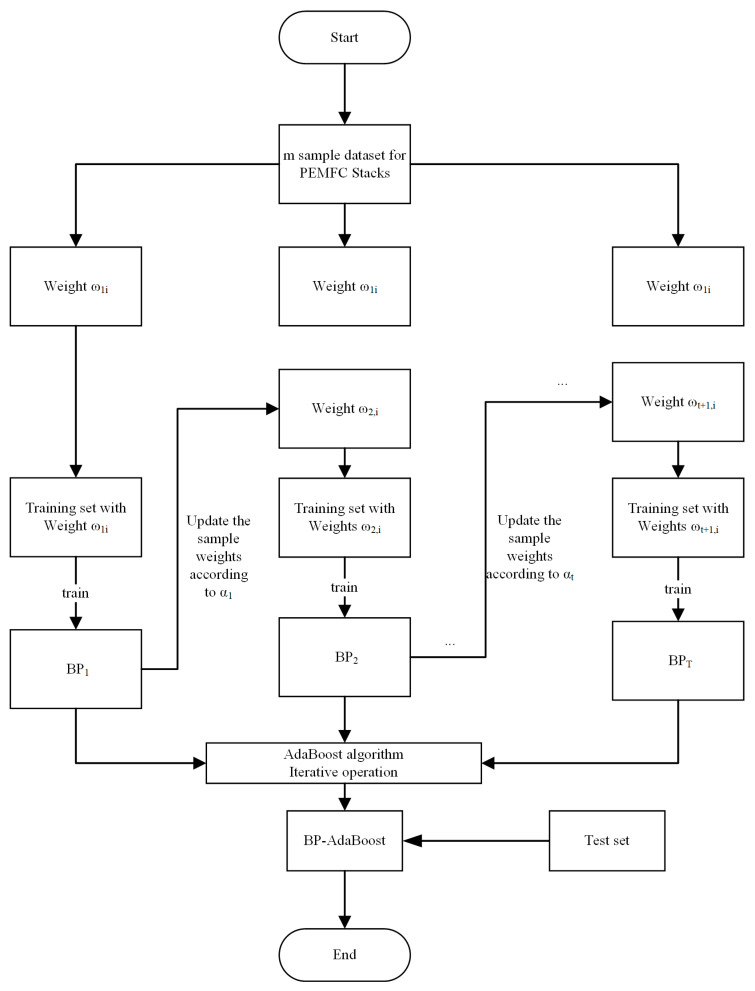
Flowchart of BP-AdaBoost algorithm.

**Figure 3 sensors-24-06120-f003:**
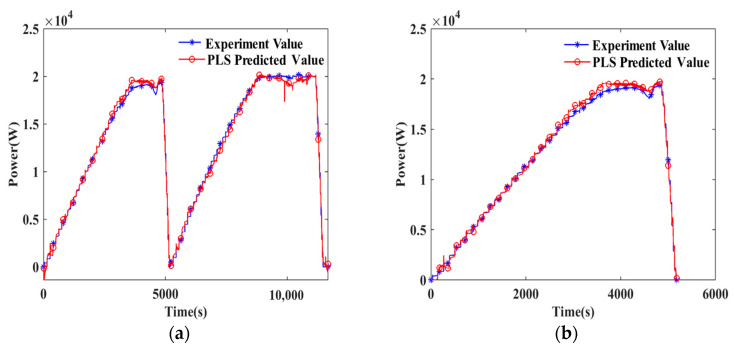
Plot of predicted power vs. actual power for the PEMFC Stack A. (1) PLS: (**a**) training set, (**b**) test set; (2) SVM: (**c**) training set, (**d**) test set; (3) BP: (**e**) training set, (**f**) test set; (4) Multiply U and I: (**g**) training set, (**h**) test set; (5) BP-AdaBoost: (**i**) training set, (**j**) test set; (6) Absolute Difference: (**k**) training set, (**l**) test set.

**Figure 4 sensors-24-06120-f004:**
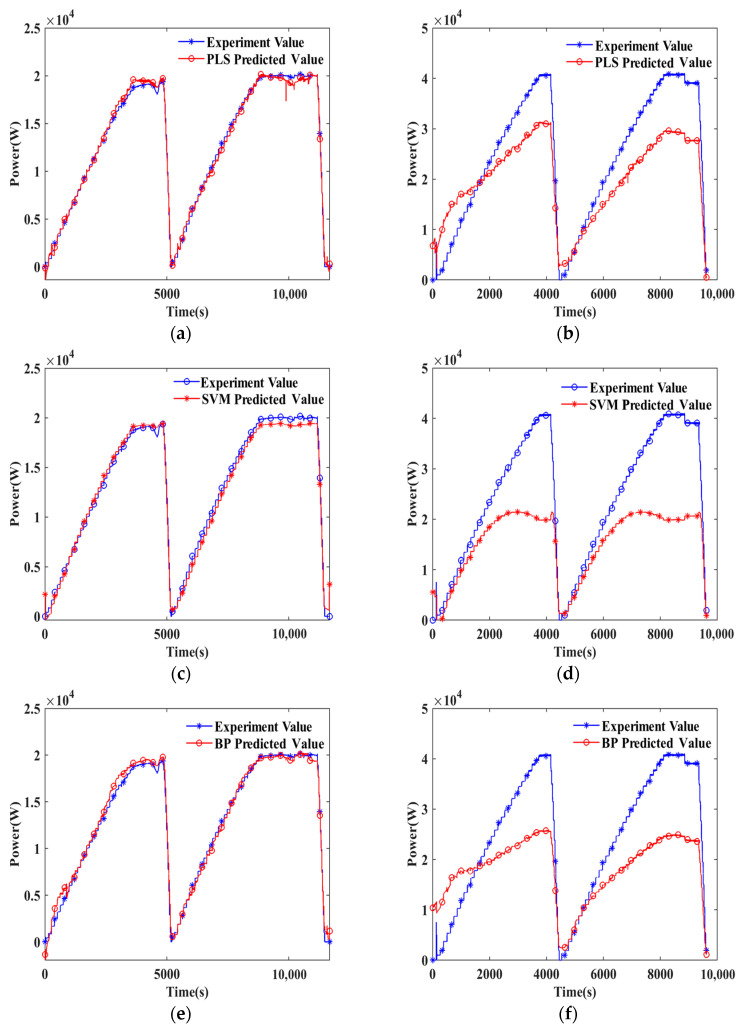
Plot of predicted power vs. actual power for PEMFC Stack B. (1) PLS: (**a**) training set, (**b**) test set; (2) SVM: (**c**) training set, (**d**) test set; (3) BP: (**e**) training set, (**f**) test set; (4) Multiply U and I: (**g**) training set, (**h**) test set; (5) BP-AdaBoost: (**i**) training set, (**j**) test set; (6) Absolute Difference: (**k**) training set, (**l**) test set.

**Table 1 sensors-24-06120-t001:** Employed Process Variables.

NO.	Variables	Unit	NO.	Variables	Unit	NO.	Variables	Unit
1	Hydrogen flow rate setting	m^3^/s	10	Air dewpoint temperature	°C	19	Air inlet temperature	°C
2	Air flow rate setting	m^3^/s	11	Hydrogen outlet pressure setting	MPa	20	Recirculating water outlet temperature	°C
3	Hydrogen inlet pressure	MPa	12	Air outlet pressure setting	MPa	21	Recirculating water inlet temperature setting	°C
4	Hydrogen outlet pressure	MPa	13	Hydrogen stoichiometry ratio setting	1	22	Recirculating water outlet temperature setting	°C
5	Hydrogen stoichiometry ratio	1	14	Air stoichiometry ratio setting	1	23	Recirculating water tank temperature setting	°C
6	Air inlet pressure	MPa	15	Recirculating water tank temperature	°C	24	Recirculating water outlet pressure	MPa
7	Air outlet pressure	MPa	16	Hydrogen outlet temperature	°C	25	Hydrogen dewpoint temperature	°C
8	Air stoichiometry ratio	1	27	Recirculating water inlet temperature	°C	26	Air outlet temperature	°C
9	Recirculating water inlet pressure	MPa	28	Recirculating water outlet temperature	°C	27	Hydrogen inlet temperature	°C

**Table 2 sensors-24-06120-t002:** Comparative study of power prediction results on PEMFC Stack A.

Method		RMSE	MAE	R^2^
PLS	training set	417.1615	346.7267	0.9959
	test set	435.9583	367.0074	0.9950
SVM	training set	586.4249	513.0359	0.9921
	test set	633.8917	515.5975	0.9896
BP	training set	561.2299	434.0151	0.99272
	test set	731.1183	602.0637	0.98618
Multiply U and I	training set	81.4409	15.6639	0.9999
	test set	83.5748	16.2860	0.9998
AdaBoost + BP	training set	188.1756	135.5977	0.9992
	test set	191.9879	137.0943	0.9991

**Table 3 sensors-24-06120-t003:** Comparative study of power prediction results on PEMFC Stack B.

Method		RMSE	MAE	R^2^
PLS	training set	417.1615	346.7267	0.9959
	test set	8281.6214	7091.9048	0.6051
SVM	training set	586.4249	513.0359	0.9921
	test set	11,509.5741	8965.7543	0.2372
BP	training set	561.2299	434.0151	0.9927
	test set	10,934.5233	8956.8967	0.3116
Multiply U and I	training set	81.4409	15.6639	0.9999
	test set	234.7260	60.8646	0.9997
AdaBoost +BP	training set	188.1756	135.5977	0.9992
	test set	7642.1355	6282.8910	0.6637

## Data Availability

The datasets generated during the current study are available from the corresponding author on reasonable request. However, due to the confidentiality agreements of the partner companies, it is not permitted to use the data publicly for any academic or commercial purposes.

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
