# Peer review of "Data-Driven Power Prediction for Proton Exchange Membrane Fuel Cell Reactor Systems"

_sensors, 2024, doi:10.3390/s24186120_

Round 1
Reviewer 1 Report
Comments and Suggestions for Authors
This paper presents a novel machine-learning method for predicting the power of hydrogen fuel cell stacks. The technique is subsequently compared to other commonly used machine learning models to demonstrate its performance. The following comments are suggested to consider.
1. probabilistic machine learning is an important branch for time-series prediction, which could be a promising trend in the power prediction field. In this case, related methods can be mentioned, such as Bayesian dynamic modelling for probabilistic prediction of pavement condition.
2. The determination of input variables for the proposed model is not clear.
3. SVM and BP are common and basic machine learning algorithms. More complicated models are suggested for comparison.
4. The hyperparameters can directly affect the model’s influence. Which method did you use to optimize the hyperparameters? Bayesian optimization, random optimization, or grid search? The optimization algorithms can refer Probabilistic framework with Bayesian optimization for predicting typhoon-induced dynamic responses of a long-span bridge.
5. The authors should ensure that each compared model has the optimal hypermeters for comparison fairness.
Comments on the Quality of English LanguageMinor editing of English language required.
Reviewer 2 Report
Comments and Suggestions for Authors
This paper is about the data-driven power prediction for proton exchange membrane fuel cell reactor systems. It introduces an innovative, cost-effective data-driven approach using the BP-AdaBoost algorithm to accurately predict the power output of hydrogen fuel cell stacks. A comparative analysis was performed to evaluate the BP-AdaBoost model against other models like the Partial Least Squares Regression (PLS) algorithm, Support Vector Machine (SVM) algorithm, and back propagation (BP) neural network, using metrics such as RMSE, MAE, and R² to assess the prediction accuracy. The results indicate that the BP-AdaBoost algorithm outperforms other models in predicting power output for electric stacks of the same type. However, in the present form, there is a requirement for many improvements:
1. For the description“A major hurdle in this field is the precise forecasting of PEMFC power output without relying on expensive Cell Voltage Monitoring systems“, is not right, cell voltage monitoring systems are a necessary component, and the readers cannot understand when the authors' description about “ it's expensive”.
2. Why is it necessary to predict the output power of fuel cells? It needs to be explained in the Introduction. The authors can refer to some literature, such as: 10.1016/j.cej.2024.151951; https://ieeexplore.ieee.org/abstract/document/10526347
3. Some data-driven methods need to be added in the Introduction and describe the advantages of data-driven methods.
For example, 'The data-driven approach avoids the complex degradation mechanism of fuel cells and uses machine learning methods to establish a black box model to complete the prediction based on the fuel cell ageing experimental data. At present, machine learning, especially deep learning, has been increasingly applied in the field of fuel cells. The common methods for the lifespan prediction of PEMFCs include long short-term memory (LSTM) neural networks, gated recurrent unit (GRU) neural networks, convolutional neural networks (CNN), and echo state networks (ESN).' There is also updated work on 'degradation prediction of proton exchange membrane fuel cell performance based on a Transformer model'.
4. How to solve the noise problem in the experiment data, which has a great impact on the accuracy of the prediction model? This issue should be further clarified.
5. Page 9, 'The primary difference between these stacks lies in the different catalysts used within their Catalyst-Coated Mem-branes (CCM). Seventy per cent of one of the Training electric stacks was used for training and the remaining 30% was used as a validation set for tuning the model parameters.'
How the model parameters are determined? Further details should be provided. See 'accurate parameter estimation of the voltage model for proton exchange membrane fuel Cells', it should be noted that to reveal the dynamic process occurring in the PEMFC, it is crucial to create a precise model to simulate and assay the output feature. However, PEMFC models are often highly nonlinear and complex, and several unidentified empirical factors are often not provided in the data sheets of the manufacturer. Hence, precise optimization techniques are needed to determine the unidentified factors that need to be concretized in the fuel cell model and to create an effective and robust electrochemical model to simulate the PEMFC features more precisely.
6. The proposed method is very common, what are the innovations of this article? The innovation points of this paper need to be further clarified.
7. The sequence numbers of the figure in the manuscript are confused, please correct them.
7. The language is poor, and the quality of English should be further improved.
Reviewer 3 Report
Comments and Suggestions for Authors
The paper, " Data-Driven Power Prediction for Proton Exchange Membrane Fuel Cell Reactor Systems" addresses an interesting topic, but needs a lot of revisions before it can be published.
1. It is necessary to rewrite the abstract. It is advised to provide some findings in the abstract.
2. Previous investigations were not adequately described in the introduction section.
3. What is the scientific gap in this field that the authors wanted to address? The originality of the research should be written more clearly at the end of the introduction section.
4. It is recommended to mention the properties of Stack A and Stack B to know what is the difference between them.
5. The numbering of the figures in the results and discussion is incorrect. It must be corrected.
6. The results and discussion section is poorly explained and the repetition is terrible. This section needs to be re-explained.
7. Very few scientific sentences and arguments can be extracted from this section as the same statement from Stack A is repeated with Stack B.
8. Figure 3 contains 12 plots, as does Figure 4, but these results are presented in a very brief and unhelpful manner.
9. English needs to be improved.
Comments on the Quality of English Language9. English needs to be improved.
Round 2
Reviewer 3 Report
Comments and Suggestions for Authors
Accept in present form.